# Measurement report: Molecular characteristics of cloud water in southern China and insights into aqueous-phase processes from Fourier Transform Ion Cyclotron Resonance Mass Spectrometry

Wei Sun[1, 2, 3], Yuzhen Fu[1, 2, 3], Guohua Zhang[1, 2, 4], Yuxiang Yang[1, 2, 3], Feng Jiang[1, 2, 3, a], Xiufeng Lian[1, 2, 3, b], Bin Jiang[1, 2], Yuhong Liao[1, 2], Xinhui Bi[1, 2, 4], Duohong Chen[5], Jianmin Chen[6], Xinming Wang[1, 2, 4], Jie Ou[7], Ping'an Peng[1, 2], Guoying Sheng[1, 2]

[1] State Key Laboratory of Organic Geochemistry and Guangdong Key Laboratory of Environmental Protection and Resources Utilization, Guangzhou Institute of Geochemistry, Chinese Academy of Sciences, Guangzhou, 510640, PR China
[2] CAS Center for Excellence in Deep Earth Science, Guangzhou, 510640, PR China
[3] University of Chinese Academy of Sciences, Beijing, 100049, PR China
[4] Guangdong-Hong Kong-Macao Joint Laboratory for Environmental Pollution and Control, Guangzhou Institute of Geochemistry, Chinese Academy of Sciences, Guangzhou 510640, PR China
[5] State Environmental Protection Key Laboratory of Regional Air Quality Monitoring, Guangdong Environmental Monitoring Center, Guangzhou 510308, PR China
[6] Shanghai Key Laboratory of Atmospheric Particle Pollution and Prevention, Department of Environmental Science and Engineering, Fudan University, Shanghai 200433, PR China
[7] Shaoguan Environmental Monitoring Center, Shaoguan 512026, PR China
[a] now at: Institute of Meteorology and Climate Research, Karlsruhe Institute of Technology, Eggenstein-Leopoldshafen 76344, Germany
[b] now at: Institute of Mass Spectrometry and Atmospheric Environment, Guangdong Provincial Engineering Research Center for On-line Source Apportionment System of Air Pollution, Jinan University, Guangzhou 510632, PR China

*Correspondence to*: Xinhui Bi (bixh@gig.ac.cn)

**Abstract.** Characterizing the molecular composition of cloud water could provide unique insight into aqueous chemistry. Field measurement was conducted at Mt. Tianjing in southern China in May, 2018. Thousands of formulas ($C_{5-30}H_{4-55}O_{1-15}N_{0-2}S_{0-2}$) were identified in cloud water by Fourier Transform Ion Cyclotron Resonance Mass Spectrometry (FT-ICR MS). CHON (formulas containing C, H, O, and N elements, the same is true for CHO and CHOS) represents the dominant component (43.6-65.3% of relative abundance), followed by CHO (13.8-52.1%). S-containing formulas constitute ~5-20% of all assigned formulas. Cloud water has relative-abundance-weighted average O/C of 0.45-0.56 and the double-bond equivalent of 5.10-5.70. Most of the formulas (> 85%) are assigned as aliphatic and olefinic species. No statistical difference of oxidation state is observed between cloud water and interstitial $PM_{2.5}$. CHON with aromatic structures are abundant in cloud water, suggesting their enhanced in-cloud formation. Other organics in cloud water are mainly from biomass burning and oxidation of biogenic volatile organic compounds. The cloud water contains more abundant CHON and CHOS at night, which are primarily contributed by $-N_2O_5$ function and organosulfates, demonstrating the enhanced formation in dark aqueous or multi-phase reactions. While more abundant CHO is observed during the daytime, likely due to the photochemical oxidation and photolysis of N-/S-containing formulas. The results provide an improved understanding of the in-cloud aqueous-phase reactions.

## 1 Introduction

On average, approximately 70% of the Earth is covered by clouds (Stubenrauch et al., 2013; Herrmann et al., 2015). Cloud water is an essential sink of organics (Herckes et al., 2013) and provides a medium for the aqueous-phase reactions of dissolved gases and aerosols (Blando and Turpin, 2000), which can substantially modify the characteristics of the organics (McNeill, 2015; Kim et al., 2019). Aqueous-phase secondary organic aerosols (aqSOA) forming in the in-cloud aqueous-phase processes significantly contribute to the total SOA with a negative impact on the visibility, human health, and climate (Ge et al., 2012; Huang et al., 2018; Schurman et al., 2018; Li et al., 2020a; Smith et al., 2014; Paglione et al., 2020; Hallquist et al., 2009). Therefore, understanding the molecular characteristics and aqueous-phase reactions in cloud droplets is crucial to assessing their impact accurately.

Organics in cloud water mainly include organic acids (e.g., formic, acetic, oxalic acids, and other short-chain mono and dicarboxylic acids) (Sun et al., 2016), carbonyls (e.g., formaldehyde, acetaldehyde, glyoxal, and methylglyoxal) (Ervens et al., 2013; Van Pinxteren et al., 2016), as well as some heteroatom-containing compounds such as amino acids (Bianco et al., 2016b), organonitrates, and organosulfates (Zhao et al., 2013). Some less polar organics such as n-alkanes (Herckes et al., 2002), benzene, toluene, ethylbenzene, and xylenes (Wang et al., 2020), polycyclic aromatic hydrocarbons (Herckes et al., 2002; Ehrenhauser et al., 2012), phenols, and nitrophenols (Lütke et al., 1997; Lütke et al., 1999) are also observed, though with a much lower fraction of dissolved organic materials (usually < 1%). The organics characterized using chromatographic and spectroscopic techniques only take a proportion of ~20% of all kinds of organics in cloud water (Herckes et al., 2013; Bianco et al., 2018). Ultra-high-resolution mass spectrometry such as Fourier Transform Ion Cyclotron Resonance Mass Spectrometry (FT-ICR MS) has made it possible to characterize individual molecular formulas in complex mixtures, although the selectivity of detection still exists (Cho et al., 2015; Hockaday et al., 2009). In studies using ultra-high-resolution MS to characterize cloud/fog water, formulas were mainly divided into CHO, CHON, CHOS, and CHONS based on the elemental composition, in which CHO and CHON were usually dominant (Zhao et al., 2013; Cook et al., 2017; Brege et al., 2018; Bianco et al., 2018; Bianco et al., 2019). Boone et al. (2015) observed a high fraction of N-containing formulas in cloud water compared with particles and attributed it to the aqueous-phase formation. However, a more recent study carried out in Po Valley observed more CHO formulas in cloud water, while particle samples contained more N- and S-containing formulas. The authors attribute it to the high possibility of reactions with sulfate and nitrate ions in the concentrated environment of aerosol liquid water (Brege et al., 2018). Thus more observations are needed to provide more convincing evidence of in-cloud aqueous-phase reactions.

Aqueous-phase reactions have been identified as an important source of organics in cloud water in addition to the gas-liquid and particle-liquid partition. Aqueous-phase reactions mainly include radical and non-radical reactions. Under irradiation, hydroxyl radical (OH•) is the primary radical in the atmosphere (Herrmann et al., 2010). In cloud water, the oxidation of precursors can be initiated by the hydrogen abstraction or electron transfer reaction driven by the OH•, resulting in the formation of organic acids and condensed compounds (McNeill, 2015). On the one hand, photolysis causes the

fragmentation of high-molecular-weight organic compounds, resulting in the formation of relatively low-molecular-weight compounds such as small acids, including oxalic, glyoxylic, and, in large quantity, formic and acetic acid (Renard et al., 2015; Schurman et al., 2018; Huang et al., 2018; Sun et al., 2010; Li et al., 2014; Löflund et al., 2002; Ye et al., 2020). These compounds are highly oxygenated owing to cloud processing (Brege et al., 2018; Sareen et al., 2016), as indicated by the fact that aqSOA has a higher O/C ratio (~1) than gas-phase SOA (0.3-0.5) in the atmosphere (Ervens et al., 2011). On the

other hand, photochemistry also leads to the oligomerization of organics, such as pyruvic acid, phenols, and methyl vinyl ketone under conditions relevant to deliquesced aerosols (Reed Harris et al., 2014; Renard et al., 2015; Yu et al., 2016). The oligomerization of tryptophan was also observed in synthetic cloud water (Bianco et al., 2016a). For the N- and S-containing organics, photochemistry may cause the release of inorganic nitrate and sulfate (Braman et al., 2020; Laskin et al., 2015; Bruggemann et al., 2020). The main radical in the atmosphere at night is $NO_3•$, which can form from the gas-phase reaction

between $NO_2$ and $O_3$ and enter cloud droplets. The reactions between $NO_3•$ and organics lead to the oxidation of organics or the addition of functional groups containing N atoms when the aqueous phase is concentrated and acidic (Herrmann et al., 2015; McNeill, 2015; Wang et al., 2008; Szmigielski, 2016; Rudziński and Szmigielski, 2019). Meanwhile, the radical nitration is believed to form dinitroaromatics in the aqueous phase (Kroflic et al., 2015). Non-radical reactions also lead to the formation of N-containing organics. Carbonyls can react with ammonium and amine without illumination, resulting in

the generation of imidazoles and N-containing oligomers, especially in aerosol liquid water and evaporating cloud water (De Haan et al., 2009; Kua et al., 2011; De Haan et al., 2011; De Haan et al., 2018). In addition, the nucleophilic addition of nitrate to the isoprene-derived epoxydiol can effectively form organonitrates (Darer et al., 2011). While organosulfates can form through heterogeneous and bulk particle-phase reactions (Brüggemann et al., 2020). Several formation mechanisms of organosulfates, such as acid-catalyzed ring-opening of epoxides, sulfate esterification, nucleophilic substitution of alcohols

with sulfuric acid, and sulfoxy radical reactions, have been proposed in recent years (Brüggemann et al., 2020). The non-radical reactions in the aqueous phase also include hydrolysis, hydration, Fenton reaction, and translation metals reactivity with organics, probably ozone reactivity at the gas/liquid interface, and so on (McNeill, 2015; Deguillaume et al., 2005; Herrmann et al., 2015).

    To date, only a few studies reported the molecular characteristics of cloud water using ultra-high-resolution MS, hampering

our understanding of aqueous-phase reactions on the composition of cloud water. In this study, cloud water and $PM_{2.5}$ samples at a remote mountain site were collected and analyzed by FT-ICR MS. The primary objectives of this study are to investigate the molecular characteristics and composition of the organics in cloud water, and to explore the potential influences of aqueous-phase reactions.

**2 Materials and methods**

## 2.1 Sample collection and pretreatment

A sampling campaign was carried out at an atmospheric monitoring station (112°53′56″E, 24°41′56″N, 1690 m above sea level) located in the Tianjing Mountain in southern China (Fig. S1). The site is located in a natural conservation zone far away from anthropogenic activities, which is affected mainly by local biogenic emissions and long-distance transport during the monsoon seasons.

A Caltech Active Strand Cloudwater Collector, *Version 2* (CASCC2), was used for cloud water collection (Demoz et al., 1996). Cloud events were identified using the humidity sensor and the visibility meter in a co-working ground-based counterflow virtual impactor (Model 1205, Brechtel Mfg., Inc., USA) according to the following criteria: visibility $\leq$ 3 km, relative humidity $\geq$ 95%, and no precipitation. During sampling, cloud droplets entered the collector powered by a rear fan, and condensed on a bank of Teflon strands at a flow rate of 5.8 $m^3$ $min^{-1}$ and collection efficiency of 86%. Condensed cloud water flowed into the pre-cleaned sample jar through a Teflon tube equipped at the bottom of the collector. The pH of the cloud water was measured using a pH meter. Samples were refrigerated immediately after sampling and kept until the analysis. The cloud liquid water content (LWC) during the sampling was calculated as follows (Guo et al., 2012):

$$LWC = \Delta m/(\Delta t \times \eta \times Q)$$

where $\Delta m$ represents the mass of the sample (g); $\Delta t$ represents the sampling interval (min). $\eta$ is the collection efficiency, which is regarded as 86% for cloud droplets larger than 3.5 $\mu m$; and Q is the airflow of the CASCC2, i.e., 5.8 $m^3$ $min^{-1}$. A total of 24 cloud water samples (Sample ID: CL1-24) were collected in succession during a long-duration cloud event that lasted from May 8 to May 13, 2018. To investigate the molecular characteristics of the organics and the effects of aqueous-phase processes, six samples collected from May 11 to May 12 (CL12-17, Table 1) were selected for FT-ICR MS analysis in detail since these six samples were all collected during the maintenance stage of a cloud event with stable pH and LWC (Table 1), stable meteorological conditions and no dramatic change of air masses origin (Fig. S1, S2). The sampling interval of six samples is presented in Table 1 and Fig. S2. The samples CL12, CL13, and CL14 were collected during the daytime of May 11, while the other three samples (CL15, CL16, and CL17) could be roughly regarded as nighttime samples, although CL15 was partly collected in the evening.

Quartz fiber filters (Whatman, Britain) were used to collected interstitial PM$_{2.5}$ samples at the same site. A PM$_{2.5}$ sampler (PM-PUF-300, Mingye Inc., China) with a cut size of 2.5 $\mu m$ at a flow rate of 300 L $min^{-1}$ was used for sampling. The sampling interval of PM$_{2.5}$ was roughly 24 hours. Two samples of PM$_{2.5}$ were collected within two days (P1: May 11 10:14-May 12 9:48, P2: May 12 10:15-May 13 10:15, respectively) during the investigated cloud events. The samples were stored at -20 ℃ immediately after sampling. For the laboratory analysis, 60 $cm^2$ of the sample filters were cut into pieces and soaked in ultra-clean water. Then the water-soluble organic matter in PM$_{2.5}$ was separated into the ultra-clean water by 30 min ultrasonic extraction three times, after that the extract was filtered using 0.22 $\mu m$ polytetrafluoroethylene filters. The extract was then pretreated and analyzed used the same methods with cloud water samples.

For FT-ICR MS analysis, water-soluble organic compounds in cloud water and PM$_{2.5}$ extract were isolated using solid-phase extraction (SPE) (Zhao et al., 2013; Bianco et al., 2018). The SPE cartridges (Strata-X, Phenomenex, USA) were pre-conditioned sequentially with 3 mL of isopropanol, 6 mL of acetonitrile, 6 mL of methanol containing 0.1% of formic acid, and 6 mL of ultrapure water containing 0.1% formic acid. Then, 40 mL of cloud water without pH adjustment and PM$_{2.5}$ extracts with pH adjusted to 4.5 using formic acid were added to the cartridge at a flow rate of approximately 1 mL min$^{-1}$. The inorganic salts were removed from the cartridge using 4 mL of ultrapure water containing 0.1% formic acid. Note that some low-weight organic molecules are expected to be lost in this step. The cartridges were then freeze-dried, and the analytes were eluted using 3 mL of acetonitrile/methanol/ultrapure water (45/45/10, *v:v:v*) at pH 10.4, with the pH being adjusted using ammonium hydroxide. All the solvents were HPLC-grade. Blank samples of the cloud water and PM$_{2.5}$ were processed and analyzed following the same procedure as the samples.

## 2.2 Instrumental analysis and data processing

A solariX XR FT-ICR MS instrument (Bruker Daltonik GmbH, Bremen, Germany) equipped with a 9.4-T refrigerated, actively shielded superconducting magnet (Bruker Biospin, Wissembourg, France) and a Paracell analyzer cell was used for the analysis in this study. An electrospray ionization (ESI) source (Bruker Daltonik GmbH, Bremen, Germany) at the negative ion mode was used to ionize the organics. ESI is a soft ionization technique that offers minimal fragmentation of the analytes (Mazzoleni et al., 2010). [M-H]$^{-}$ was detected at the negative ion mode. The coupling of ESI and FT-ICR MS with ultra-high mass resolution makes it possible to characterize the element constitution within molecules. Note that ESI is efficient at ionizing molecules having polar functional groups containing nitrogen and oxygen atoms (Cho et al., 2015). The direct infusion method was used in this study. The samples were redissolved in 1 mL of methanol and injected into an ESI source at a flow rate of 200 μL h$^{-1}$. A nebulizer gas pressure of 1 bar, a dry gas velocity of 4 L min$^{-1}$ and temperature of 200 ℃, and capillary voltages of +4500 V and the end plate offset -500 V were used for ESI source. The optimized mass for quadrupole (Q1) was 170 Da. An argon-filled hexapole collision pool was operated at 2 MHz and 1400 Vp-p RF amplitude. The time of flight was 0.7 ms and the mass range was 150-800 Da and the ion accumulation time was 0.1 s. A total of 128 continuous 4M data FT-ICR transients were co-added to enhance the signal-to-noise ratio and dynamic range. A typical mass-resolving power (*m*/Δ*m*50%, where Δ*m*50% is the magnitude of the mass spectral peak full width at half-maximum peak height) of more than 450,000 at *m/z* 319 with <0.3 ppm absolute mass error was achieved. The mass spectra were calibrated externally using measurements of a known homologous series of N$_1$ and O$_2$ molecules (e.g., C$_{16}$H$_{31}$O$_2$, C$_{17}$H$_{33}$O$_2$, and C$_{18}$H$_{35}$O$_2$, etc. that only separated by –CH$_2$ units) frequently detected in a crude oil sample before sample detection. The final spectrum was internally recalibrated with typical class species peaks in cloud water samples (-O$_4$ species) using quadratic calibration in DataAnalysis 5.0 (Bruker Daltonics) (Jiang et al., 2019). All the mathematically possible formulas for all the ions with a signal-to-noise ratio greater than 10, considering a mass tolerance of ±0.6 ppm were calculated. The maximum numbers of atoms for the formula calculator were set to: 30 $^{12}$C, 60 $^{1}$H, 20 $^{16}$O, 2 $^{14}$N, 2 $^{32}$S, 1 $^{13}$C, 1 $^{18}$O, and 1 $^{34}$S. Formulas assigned to isotopomers (i.e., $^{13}$C, $^{18}$O, or $^{34}$S) were not discussed in this study. Thus, the chemical formula

$C_cH_hO_oN_nS_s$ was obtained. Future selecting was applied using following criteria to exclude formulas not detected frequently in natural materials: O/C ≤ 1.2, 0.3 ≤ H/C ≤ 2.25, N/C ≤ 0.5, S/C ≤ 0.2, 2C + 2 > H, C + 2 > O and obeying N rule. Finally, only intensities of sample ion peaks enhanced at least 100 times higher than those in the blank were retained for further data analysis in order to avoid possible contamination. The double-bond equivalent (DBE) can be used to evaluate the number of rings and double bonds in a molecule (An et al., 2019). The DBE of each assigned molecular formula ($C_cH_hO_oN_nS_s$) was

calculated as follows:

$$DBE = (2c + 2 - h + n)/2$$

The oxidation state of carbon atoms ($OS_C$) was calculated as follow based on the approximation described in Kroll et al. (2011) and Brege et al. (2018):

$$OS_C \approx 2*o/c - h/c - 5*n/c - 6*s/c$$

The modified aromaticity index ($AI_{mod}$) was first proposed by Koch et al. to evaluate the aromaticity of high-resolution mass data (Koch and Dittmar, 2006):

$$AI_{mod} = (1 + c - 0.5*o - s - 0.5*h)/(c - 0.5*o - s - n)$$

$AI_{mod} \geq 0.5$ and 0.67 represent the existence of aromatic and condensed aromatic structures (Koch and Dittmar, 2006).

Relative abundance of each formula was represented by the intensities of each peak after normalization by the maximum

intensity in each sample. Note that both the recovery of SPE and the selective ionization of negative ESI might cause a bias of mass spectra to certain peaks. And ESI FT-ICR MS is not a purely quantitative technique, the intensity of the peak for each formula is a product of its concentration and ionization efficiency. However, since all of the samples were pretreated using the same procedure and measured using the same instrumental conditions, each spectrum was biased in an equal fashion, so relative peak intensities within the acquired spectra can be compared to each other, although they cannot be

related back to concentrations in the original samples (Sleighter et al., 2010; Wozniak et al., 2014). The relative-abundance-weighted average elemental ratios of oxygen, carbon, and hydrogen (i.e., O/C, H/C, etc.) and other characteristic parameters were calculated following Song et al. (2018):

$$O/C_w = \Sigma(O/C_i \times Int_i)/\Sigma Int_i$$

$$H/C_w = \Sigma(H/C_i \times Int_i)/\Sigma Int_i$$

$$DBE_w = \Sigma(DBE_i \times Int_i)/\Sigma Int_i$$

$$OS_{Cw} = \Sigma(OS_{Ci} \times Int_i)/\Sigma Int_i$$

where $Int_i$ represents the intensity of the mass spectrum for each individual molecular formula, $i$. The discussion in this paper is based on the weighted average values unless otherwise specified; thus, the subscript "w" is omitted for brevity in the following texts.

Cloud water samples were also analyzed for ionic species. Descriptions of these analyses are available elsewhere (Guo et al., 2012; Bianco et al., 2018). Briefly, water-soluble inorganic ions and oxalate ($C_2O_4^{2-}$) were detected using an Ion Chromatograph (883 Basic IC plus, Metrohm, Switzerland) with suppressed conductivity detection and a Metrosep C4-150/4.0 column (Metrohm) for cations and a Metrosep A Supp 5-150/4.0 column (Metrohm) for anions.

The 72 h back trajectories were displayed using the Hybrid Single-Particle Lagrangian Integrated Trajectory model (https://ready.arl.noaa.gov) (Stein et al., 2015; Lin et al., 2017). The endpoint of the trajectory in the model was set to a height of 1800 m above sea level. In addition, the meteorological conditions during sampling and water-soluble ions concentrations are provided and discussed in Text S1.

## 3. Results and Discussion

### 3.1 Overview of molecular formulas of cloud water and comparison to the interstitial $PM_{2.5}$

A total of 1691, 1546, 1604, 1264, 2364, and 2767 molecular formulas were identified in CL12-17 samples, respectively. According to the elemental compositions, four groups (CHO, CHON, CHOS, and CHONS) were assigned. The reconstructed mass spectrum of a typical sample, CL12, is presented in Fig. 1a. The most intensive ion peaks are within the range of $m/z$ 200-400. A similar pattern is also found in cloud (Zhao et al., 2013; Bianco et al., 2018; Cook et al., 2017), fog (Brege et al., 2018), and aerosols (Lin et al., 2012; Mazzoleni et al., 2012).

In cloud water, CHON is the most frequently observed group, representing more than 60% of the total number of assigned formulas. CHO contributes to 16.3-28.3% of the total number of identified formulas, while the proportions of S-containing formulas (CHOS and CHONS) are much lower (3.6-9.4% and 3.7-9.3%, respectively) (Table S2). The relative abundance of each group is evaluated, as shown in Fig. 1b and Table S2. The fraction in relative abundance ($f_{RA}$) of the CHON group is 43.6-65.3% (54.9% on arithmetic average), and CHO contributes 13.8-52.1% (34.7% on arithmetic average). S-containing formulas constitute the remaining fraction, approximately 5-20% (Table S2). The fractions of four groups in relative abundance and in number are different, which is mainly attributed to some formulas with high intensities contributing much more to relative abundance than to the number. Note that the abundant CHO and CHON cannot be directly related back to the composition of samples since the preferential detection of these molecules in negative ESI. However, the comparison among the samples is still meaningful since they are expected to have the same bias.

The cloud water shows a distinct pattern of molecular composition with the interstitial $PM_{2.5}$. In two $PM_{2.5}$ samples, 1198 and 1057 formulas are identified, in which CHO and CHON are dominant. The smaller number of assigned formulas in $PM_{2.5}$ may be mainly related to the low concentration of total organics in $PM_{2.5}$ extracts. The CHO group contributes to 39.9-49.8%, while the CHON group contributes to 31.8-51.0%. The S-containing formulas constitute the remaining fraction (9.1-18.4%). Similar results can also be obtained by $f_{RA}$ of CHO (44.0-55.5%), CHON (24.3-47.3%), and S-containing formulas (8.7-20.3%) (Table S3). At three sites in Pearl River Delta (PRD), more than half of all detected formulas were assigned as CHO, while CHON only accounted for 8-19% in the aerosols (Lin et al., 2012). The higher fraction of CHON in cloud water compared with $PM_{2.5}$ in PRD and interstitial particles is consistent with the previous finding (Boone et al., 2015), likely indicating the formation of N-containing organics in cloud water.

## 3.2 Effects of cloud processing on oxidation metrics and aromaticity of the molecular formulas

**Oxidation metrics** O/C and $OS_C$ are employed to evaluate the oxidation degree of molecules in cloud water. In six cloud water samples, the average O/C values range from 0.45 to 0.56 (Table S4). No significant differences of O/C are observed between cloud water and $PM_{2.5}$, of which average O/C values are 0.45-0.56 (Table S5). In the Van Krevelen (VK) plot, CHON formulas distribute in a wide area (Fig. 2 and Fig. S3). Some of them with O/C exceeding 0.8 distribute in the top-right corner of the plots, which also result in higher O/C of CHON on average (0.51-0.62, Table S4) compared with CHO, of

which O/C ratios range from 0.34 to 0.46. On relative-abundance-weighted average, the O/C ratios of CHOS and CHONS range from 0.36 to 0.51 and from 0.65 to 0.88, respectively. This is not unintelligible if we note that the N and S atoms in the CHON and S-containing formulas would probably combine with O (e.g., $-NO_2$, $-NO_3$, or $-SO_3$ function group), leading to the higher average O/C of non-CHO formulas. The $OS_C$ value excludes the influence of oxygen atoms combined with H, N, and S; thus it is a more applicative proxy to evaluate the oxidation state of carbon atoms. The average $OS_C$ values range from -

0.91 to -0.72 in cloud water (Table S4), while that in $PM_{2.5}$ ranges from -0.84 to -0.61 (Table S5). Being limited by the sampling size, no statistical difference of $OS_C$ can be identified between cloud water and $PM_{2.5}$. However, a higher $OS_C$ of detected formulas, especially CHO, appears in $PM_{2.5}$ samples (-0.40 in P2 sample). A similar phenomenon is also observed in CHOS. This is not consistent with the current understanding that precursors and products in the aqueous phase have a higher O/C, which generally causes the high water-solubility of molecules (Ervens et al., 2011). However, a previous aircraft

sampling also observed a lower O/C in cloud water than below-cloud atmospheric particles (Boone et al., 2015), suggesting that the effects of aqueous-phase reaction may be complex in the actual atmosphere. Previous studies using the large-eddy simulation model have shown that the in-cloud residence time of the parcel is on the scale of a few minutes (Stevens et al., 1996; Feingold et al., 1998), thus some masses formed in cloud droplets may remain in aerosols via the evaporation of the droplets, resulting some high oxidation organics entering the interstitial $PM_{2.5}$. We cannot completely rule out the influence

of cloud cycling, however, this impact may be limited because of the stable meteorological conditions with constant temperature, wind and saturated or supersaturated water vapor during sampling (Fig. S2). We note that $PM_{2.5}$ samples were collected during the cloud event; high aerosol liquid water content in $PM_{2.5}$ likely provides a sink containing more concentrated precursors for the aqueous-phase reactions compared with cloud water. However, no formation mechanism of more oxidized organics in aerosol liquid water is proposed in previous studies; thus future researches of aqueous-phase

reactions in the atmosphere are needed. In four groups, CHO has the highest $OS_C$ values (-0.80 ~ -0.54) (Table S4), which may be related to the high abundance of carboxyl groups in CHO.

   To investigate the diurnal variation of oxidation metrics in cloud water, daytime and nighttime samples are compared. The O/C ratios show no identified diurnal variation except for CHO groups. The O/C ratios and $OS_C$ of CHO collected during the daytime is slightly lower than the nighttime (Table S4), this is not consistent with the high oxidation capacity under the

illumination during the daytime, indicating that the oxidation degree of the organics in cloud water is not exclusively

affected by the illumination. The difference of air masses' origin and the aging processes may also influence the cloud chemistry. However, since the database is limited, the further conclusion cannot be drawn based on them.

**Aromaticity** The unsaturation and aromaticity of molecular formulas can be evaluated using the H/C ratios and the DBE, where low H/C and high DBE indicate a high degree of unsaturation, and to some extent, aromatic structure. On relative-abundance-weighted average, H/C ratios in cloud water ranges from 1.44 to 1.49, with no statistical difference than $PM_{2.5}$ (1.40-1.53). In the VK plot, CHOS and CHONS occupy an upper area of the diagram with high H/C ratios, indicating that they may have higher saturation compared with CHO and CHON. On weighted average, DBE values in cloud water range from 5.10 to 5.70 (Table S4), which is generally higher than that in $PM_{2.5}$ (4.74-5.04, Table S5). The weighted average DBE values of CHO, CHON, CHOS, and CHONS are 4.96-6.12, 5.44-6.09, 2.72-4.58, and 3.01-4.29, respectively. DBE values are also projected onto the plots of DBE versus carbon atom numbers (Fig. 3). DBE values generally increase with carbon number, and CHOS and CHONS distribute in a range with low DBE values. The higher unsaturation degree of CHO and CHON is likely corresponding to the high abundance of aromatic functions.

Another commonly used metric of aromaticity is $AI_{mod}$. In cloud water, most of the formulas (91.1-98.3% of CHO, 79.2-97.5% of CHON, 93.5-98.8% of CHOS, and 95.7-100.0% of CHONS in $f_{RA}$) are assigned as aliphatic or olefinic molecules (Table S6). The $f_{RA}$ of aliphatic and olefinic molecules in $PM_{2.5}$ also exceeds 90%. The high fraction of aliphatic and olefinic and the low fraction of aromatic structure are also observed in aerosols at a remote site (An et al., 2019). However, it is quite different from the primary emissions, including biomass burning, coal combustion, and traffic emission, of which the fraction of aromatic structures is higher (Song et al., 2018; Tang et al., 2020). The urban aerosols collected in Guangzhou, southern China, which may be mainly influenced by local primary emissions, also have a high fraction of aromatic molecules (> 20%) (Zou et al., 2020), implying the aging processes likely reduce the aromaticity of organics. In four groups of molecules in cloud water, CHON has the most (2.5-20.8% in $f_{RA}$) aromatic structures, consistent with the high DBE values (unsaturation) of CHON. Previous studies conducted in the Po Valley, Italy (Brege et al., 2018) and Fresno, USA (Leclair et al., 2012) also observed a higher fraction of aromatics in CHON than the S-containing groups in fog water. The aromatic species may provide the precursors of aqueous-phase reactions (Wang et al., 2021). While in this study, the possible dinitrophenols in cloud water contribute to the high $f_{RA}$ of aromatic structures in CHON significantly, which may be related to the aqueous-phase reactions (see the detailed discussion in Section 3.4).

### 3.3 Molecular composition of cloud water

**CHON** In cloud water, CHON formulas show no prominent carbon number peaks except sample CL17 (Fig. S4), and one or two nitrogen atoms are assigned to them (Fig. 4). Both the $N_1$ and $N_2$ categories contain 1-14 oxygen atoms. The most abundant class of $N_1$ formulas is $-N_1O_8$ or $-N_1O_7$ (Fig. 4), which include $C_{12}H_{17}NO_8$, $C_{15}H_{19}NO_8$, $C_{17}H_{27}NO_7$, $C_8H_{11}NO_7$, and so on. More than 77.7% of the CHON formulas in $f_{RA}$ in all six samples have O/N ratios exceeding 3, indicating that the N atoms in these molecules may be in the $-NO_3$ functional group (Zhao et al., 2013). Samples CL12-16 show no prominent peak of the function classes in the $N_2$ category, but a dominant peak of the $-N_2O_5$ class is observed in CL17 (Fig. 4), where

$C_8H_8N_2O_5$ and $C_7H_6N_2O_5$ are the two most abundant formulas. These formulas probably belong to dinitrophenols and their derivatives.

To evaluate the contribution of primary sources, we compared the molecular composition in cloud water with that in particles emitted from the primary sources such as biomass burning (including corn straw, pine branches, and rice straw) and coal combustion using the same analytical instrument (Song et al., 2018). In cloud water, 40.9-51.4%, 21.9-27.1%, and 48.1-59.4% (in terms of number fraction) of CHON molecules appear in the smoke particles of corn straw, pine branches, and rice straw, indicating a non-negligible contribution from biomass burning. While only 7.7-10.5% of molecules in cloud water can be corresponding to the coal combustion emission, suggesting its less contribution to the molecular composition of cloud water. Note that the comparison was only based on the molecular formulas given by FT-ICR MS, the isomeride could not be distinguished; thus the results only represented a possible relationship with the different sources. Additionally, some N-containing molecules are also detected in monoterpene SOA (Park et al., 2017; Zhang et al., 2018), in which the products such as $C_7H_{9,11}NO_{7-8}$, $C_8H_{11}NO_{7-8}$, $C_9H_{13,15}NO_{7-8}$, and $C_{10}H_{15,17,19}NO_{7-8}$ were detected in cloud water, indicating a contribution from monoterpene oxidation.

**CHO** For CHO formulas in cloud water, a prominent $C_{17}$ peak is observed in all six samples (Fig. S4), in which the most abundant formula is $C_{17}H_{26}O_4$, which also causes a significant peak of $O_4$ class (Fig. 4). The formula may belong to lipids-like molecule based on the classification in the VK plot (Bianco et al., 2018). Considering the intensive emission of biogenic volatile organic compounds (BVOCs) around the sampling site, the oxidation products of BVOCs may also contribute to the molecular composition. Putman et al. (2012) reported the molecular composition of α-pinene ozonolysis SOA using FT-ICR MS. Including the most abundant $C_{17}H_{26}O_4$, 24.0-39.1% (in term of number fraction) of the CHO formulas in cloud water are corresponding to the yields of α-pinene ozonolysis. In addition, cloud water contains formulas that have been observed in the combustion of coal and biogenic materials. Specifically, 24.2-35.8%, 65.9-77.3%, 50.7-69.3%, and 61.1-76.7% of the CHO formulas in cloud water, by number, were detected in the smoke emitted from the combustion of coal, corn straw, pine branches, and rice straw, respectively (Song et al., 2018), indicating that combustion also potentially contributes to the CHO group in cloud water.

**S-containing formulas** Most of the CHOS formulas in cloud water have $C_{13}$ or $C_{14}$ peaks (Fig. S4), and $-SO_3$ or $-SO_4$ represent the most abundant classes of CHOS (Fig. 4). We divided the CHOS formulas into two classes according to the O/S ratios: $CHOS_{O/S \geq 4}$ and $CHOS_{O/S<4}$. When O/S ≥ 4, CHOS can be provisionally identified as organosulfates (Lin et al., 2012). The O/S ratios of most of the CHOS formulas (78.9-95.8% in number fraction and 87.5-98.6% in $f_{RA}$) in cloud water exceed 4. $CHOS_{O/S<4}$ accounts for 1.4-12.5% in $f_{RA}$ of all the CHOS formulas in cloud water, indicating reduced S groups exist in these formulas. Some of them are aliphatic-like, such as $C_{24}H_{42}O_3S$, $C_{29}H_{52}O_3S$. Some are aromatic-like with high DBE values, such as $C_6H_6O_3S$, which may be an aromatic ring bearing a $-SO_3H$ group. Some of them may have more than one aromatic ring, such as $C_{17}H_{16}O_3S$, $C_{20}H_{18}O_3S$. Note that the aromaticity of these formulas cannot be identified accurately using $AI_{mod}$ values since the value is a conservative method to evaluate the aromaticity (Koch and Dittmar, 2006). The presence of aromatic structure in these molecules indicates that they are likely emitted by anthropogenic sources or biomass

burning (Ervens et al., 2011). Most of the CHONS formulas clearly peak at $C_{10}$ (Fig. S4) and have more than seven O atoms (Fig. 4), allowing the presence of both sulfate and nitrate functional groups. These species can be nitrooxy organosulfates, which have been widely observed in the cloud/fog water (Zhao et al., 2013; Brege et al., 2018) and aerosols (Wozniak et al., 2014). For the detected S-containing formulas in cloud water, 6.2-23.0%, 15.9-33.6%, and 15.0-34.3% in terms of number fraction are corresponding to the molecules in particles emitted by burning of corn straw, pine branches, and rice straw, respectively, while 21.2-43.1% (37.3% on arithmetic mean) are corresponding to that in coal combustion (Song et al., 2018), indicating coal combustion contributes to S-containing formulas in cloud water more significantly compared with that to CHO and CHON.

## 3.4 Organic matter formed by in-cloud aqueous-phase reactions

### 3.4.1 Formation of dinitrophenols

To investigate the formation of molecules in cloud water, we compared the molecular formulas in cloud water with $PM_{2.5}$. For CHON, the $-N_1O_8$ or $-N_1O_7$ formulas are also abundant in $PM_{2.5}$ samples (Fig. S5), suggesting that these formulas may not only form in cloud water. However, the formulas with high intensities, e.g., $C_8H_8N_2O_5$, $C_7H_7N_2O_5$, and $C_6H_4N_2O_5$ in cloud water, are not detected in $PM_{2.5}$ samples. Earlier studies have found that over one-third of the nitrophenols and the majority of the dinitrophenols are contributed by secondary formation (Harrison et al., 2005). The transforming from 2-nitrophenol into 2,4-dinitrophenol was also observed during cloud events (Lüttke et al., 1997; Lüttke et al., 1999). Aqueous-phase radical nitration of mononitroaromatics has been reported to be a potential pathway to form dinitroaromatics (Lüttke et al., 1999; Kroflic et al., 2015; Cook et al., 2017). This implies that in-cloud aqueous-phase reactions represent the main formation pathway of dinitrophenols at the observation site.

Generally, CHON in cloud water has a higher $f_{RA}$ during the nighttime (56.2-65.3%) compared with the daytime (43.6-54.9%) (Table S2), which is consistent with the previous findings for aerosols (O'brien et al., 2014). Particularly, the relative abundance of possible dinitrophenols formulas increases significantly at night. The representative formulas including $C_6H_4N_2O_5$, $C_7H_6N_2O_5$, and $C_8H_8N_2O_5$ account for 0.5%, 0.1%, 0.4%, 0.7%, 2.1%, and 14.8% of CHON in $f_{RA}$ for samples CL12-17, respectively. Previous studies have revealed the differences in atmospheric chemistry between day and night. The daytime chemistry is dominant by photochemical reactions, in which OH radical oxidation and photolysis represent the main processes in the aqueous phase (Ervens et al., 2011). While during the nighttime, $NO_3$ radical is dominant (Herrmann et al., 2010). The radical nitration of phenols by $NO_2$ and $NO_3$ radicals leads to the formation of nitrophenols (Harrison et al., 2005). Thus the high abundance of $-N_2O_5$ formulas may attribute to the aqueous-phase formation of these possible dinitrophenols at night. While during the daytime, the direct photolysis of nitrophenols would release $NO_2^-$ and $NO_3^-$ (Harrison et al., 2005; Chen et al., 2005; Bejan et al., 2006), causing the observed low relative abundance of dinitrophenols. A recent study conducted at the Field Museum Tama Hill, Japan observed that aerosol liquid water accelerated the formation of water-soluble organic nitrogen (WSON), especially at night, and the authors suggested that aqueous-phase reactions

between $NH_4^+$/reactive nitrogen and BVOCs at night contribute significantly to WSON in particles (Xu et al., 2020). While in this study, the elevated abundance of N-containing organics in cloud water at night is mainly contributed by dinitrophenols and their derivatives, which are the products of radical nitration in the aqueous phase, indicating another possible pathway for the generation of WSON.

### 3.4.2 Formation of oxygenated organic matter and organosulfates

For CHO, the most abundant $C_{17}H_{26}O_4$ in cloud water was also detected in α-pinene ozonolysis SOA as we discussed in Section 3.3. However, it was not detected in $PM_{2.5}$ in this study, indicating that it may mainly form through in-cloud aqueous-phase reactions. Interestingly, CHO formulas in $PM_{2.5}$ samples peak at $O_8$ (Fig. S5), which is significantly higher than cloud water (Fig. 4). This is consistent with the higher $OS_C$ values appearing in $PM_{2.5}$ samples. Some highly oxidized molecules (HOMs, O/C ≥ 0.6), e.g., $C_7H_{10}O_5$, $C_8H_{12}O_5$, $C_{13}H_{24}O_{13}$, are identified in cloud water. However, the HOMs in
cloud water only account for 12.6-32.2% of the total CHO in terms of $f_{RA}$. A higher $f_{RA}$ of CHO is observed during the daytime (Fig. 1b), which may result from the photochemical oxidation (e.g., the oxidation of volatile organic compounds) (Ehn et al., 2014; Wang et al., 2017) and photolysis of N- and S-containing formulas in cloud water under the illumination (Brüggemann et al., 2020; Laskin et al., 2015).

  For CHOS formulas, the most abundant functions classes are similar between cloud water and $PM_{2.5}$. No statistical
difference of the fraction of organosulfates is observed between cloud water and $PM_{2.5}$ except for a low $f_{RA}$ (69.5%) of organosulfates in P2 sample, which may indicate the wide variety of formation mechanisms (e.g., acid-catalyzed particle-phase reactions, nucleophilic substitution reactions in aqueous phase) and/or other common sources of CHOS in cloud water and $PM_{2.5}$ (Brüggemann et al., 2020), but possible slightly enhanced formation of that in cloud water. S-containing formulas in cloud water are abundant at night (8.4-21.0% in $f_{RA}$) compared with daytime (4.3-10.2% in $f_{RA}$). We note that the $f_{RA}$ of
$CHOS_{O/S≥4}$ at night (92.9-98.6%) is slightly higher than that during the daytime (87.5-92.2%). Thus the formation of organosulfates likely enhances at night. In contrast, the photochemical oxidation of organosulfates results in the release of inorganic sulfate during the daytime, causing a low fraction of organosulfates.

### 4 Conclusions and Atmospheric Implications

  This study investigated the molecular characteristics of cloud water using ESI FT-ICR MS and highlighted the crucial effects
of in-cloud aqueous-phase reactions on the molecular composition and characteristics of cloud water. Thousands of formulas, including CHO, CHON, CHOS, and CHONS, were detected, in which CHON and CHO formulas are dominant. Previous studies expected a higher oxidation state of organics in cloud water. However, no statistical difference between cloud water and $PM_{2.5}$ is observed in this study. While a higher $OS_C$ of detected formulas, especially CHO, appears in $PM_{2.5}$ samples. Most formulas are identified as aliphatic and olefinic species, CHON and their aromatic structures are abundant in cloud
water.

Our results showed that N-containing formulas are the most abundant in cloud water, which may mainly relate to the aqueous-phase formation. Dinitrophenols and derivatives exist abundantly in cloud water, especially at night, suggesting the contribution of radical nitration on N-containing organics in cloud water. Meanwhile, organosulfates are also detected in cloud water, and a slightly higher fraction is observed at night, suggesting the dark-reaction formation. Nitroaromatic compounds have been identified as one of the major light absorption components in brown carbon (Li et al., 2020b) and regarded as the phytotoxin as well as suspected carcinogenic materials (Harrison et al., 2005). Organosulfates are thought to affect the physicochemical properties of aerosol, such as hygroscopicity and cloud condensation nuclei formation potential (Brüggemann et al., 2020). Thus the aqueous-phase formation of N-containing organics and organosulfates at night are worth targeting. We noted that the database for the diurnal variation analysis is limited in this study, but the results provided novel insights into the diurnal variation of cloud chemistry. Firm conclusions warrant future field studies.

*Supplement.* Supporting information includes one text (Text S1), five figures (Fig. S1-S5), and six tables (Table S1-S6) related to the manuscript.

*Data availability.* The raw data of this study can be obtained by contacting the corresponding author.

*Author contributions.* XB and GZ designed the research with input from XW, PP and GS. YF, FJ and YY collected samples. WS, BJ carried out the sample pretreatment and instrumental analysis under the guidance of YL. WS processed data when YF and XL gave constructive discussion. WS wrote the manuscript, and XB, GZ and YF interpreted data and edited the manuscript. JC, DC, and JO had an active role in supporting the sampling work. All authors contributed to the discussions of the results and refinement of the manuscript.

*Competing interests.* The authors declare that they have no conflict of interest.

*Acknowledgements.* The authors gratefully acknowledge Jianzhong Song and Chunlin Zou (Guangzhou Institute of Geochemistry, Chinese Academy of Sciences) for the guidance and assistance during sample pretreatment and providing the raw data related to the article *Song et al., 2018*, which is helpful to the discussion of this manuscript.

*Financial support.* This work was supported by National Nature Science Foundation of China (41877307 and 42077322), Natural Science Foundation of Guangdong Province (2019B151502022), and Guangdong Foundation for Program of Science and Technology Research (2019B121202002 and 2019B121205006).

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

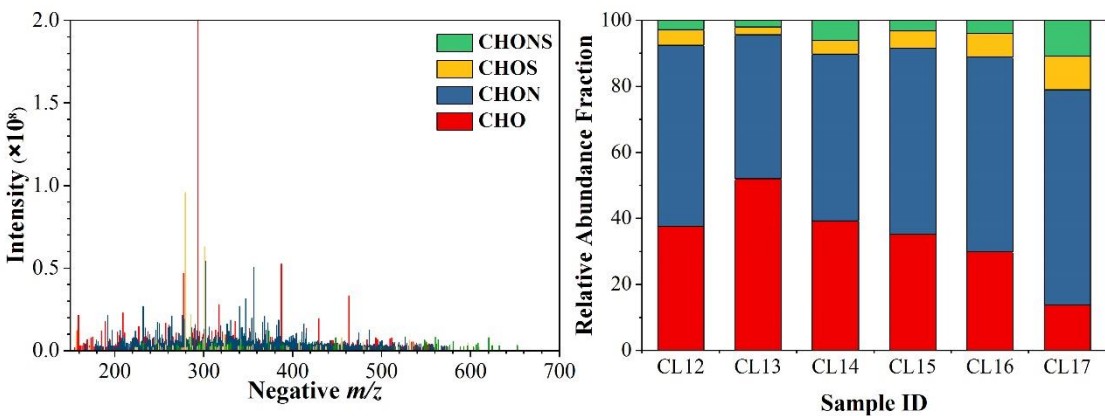

**Figure 1. Reconstructed FT-ICR mass spectra of a typical sample, CL12 (a); Relative abundance fraction of the four groups (CHO, CHON, CHOS, and CHONS) in the six cloud water samples (b).**

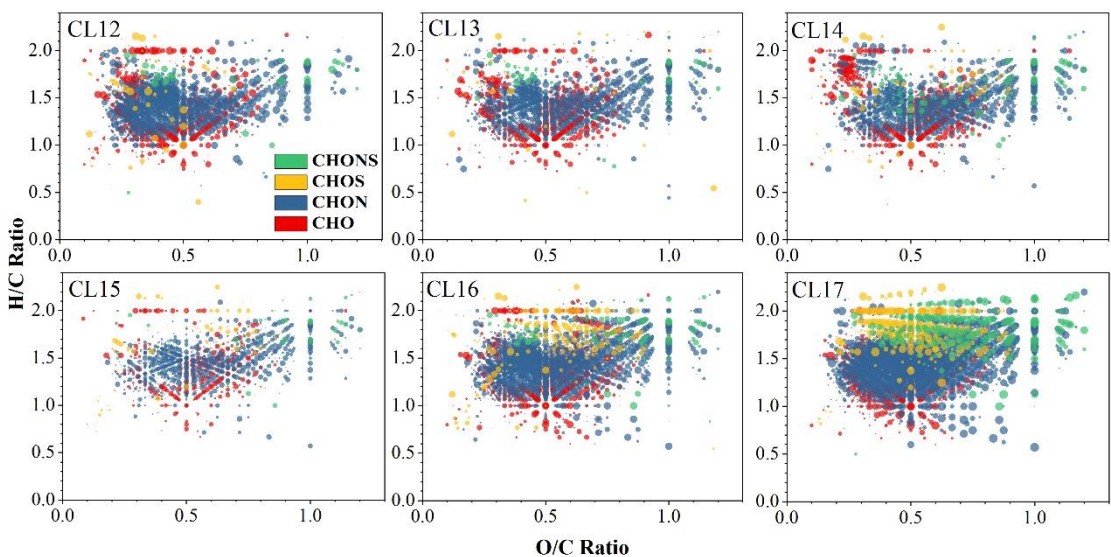

**Figure 2. Van Krevelen diagrams as a function of four groups (CHO, CHNO, CHOS, and CHNOS) for the cloud water samples.**
**The larger point in the diagram represents the higher relative abundance of the formula.**

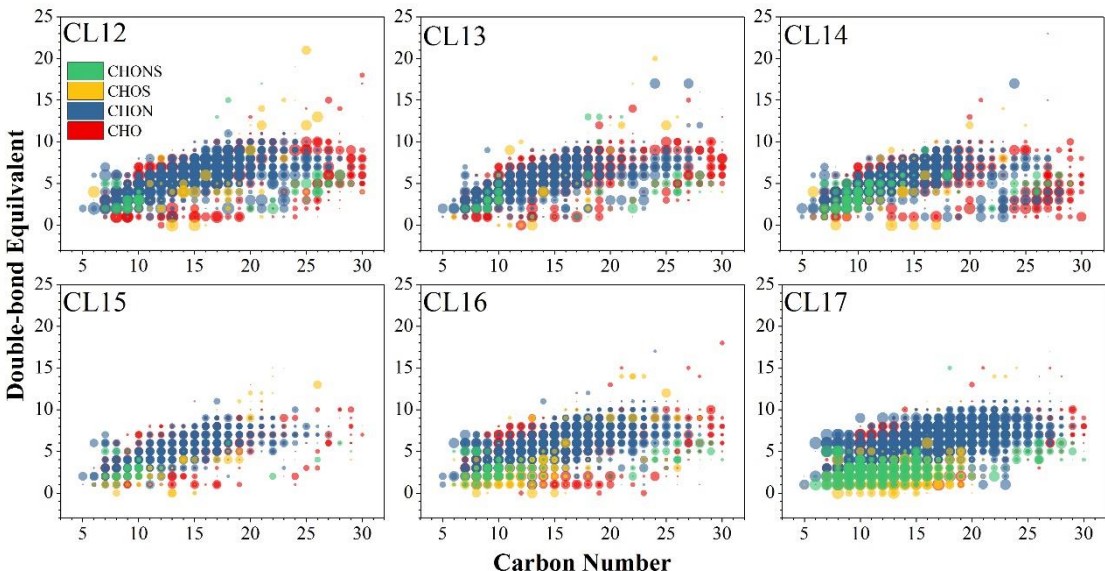

**Figure 3. The double bond equivalent (DBE) versus the number of C atoms for unique molecular formulas in cloud water samples. The larger point in the diagram represents the higher relative abundance of the formula.**

670

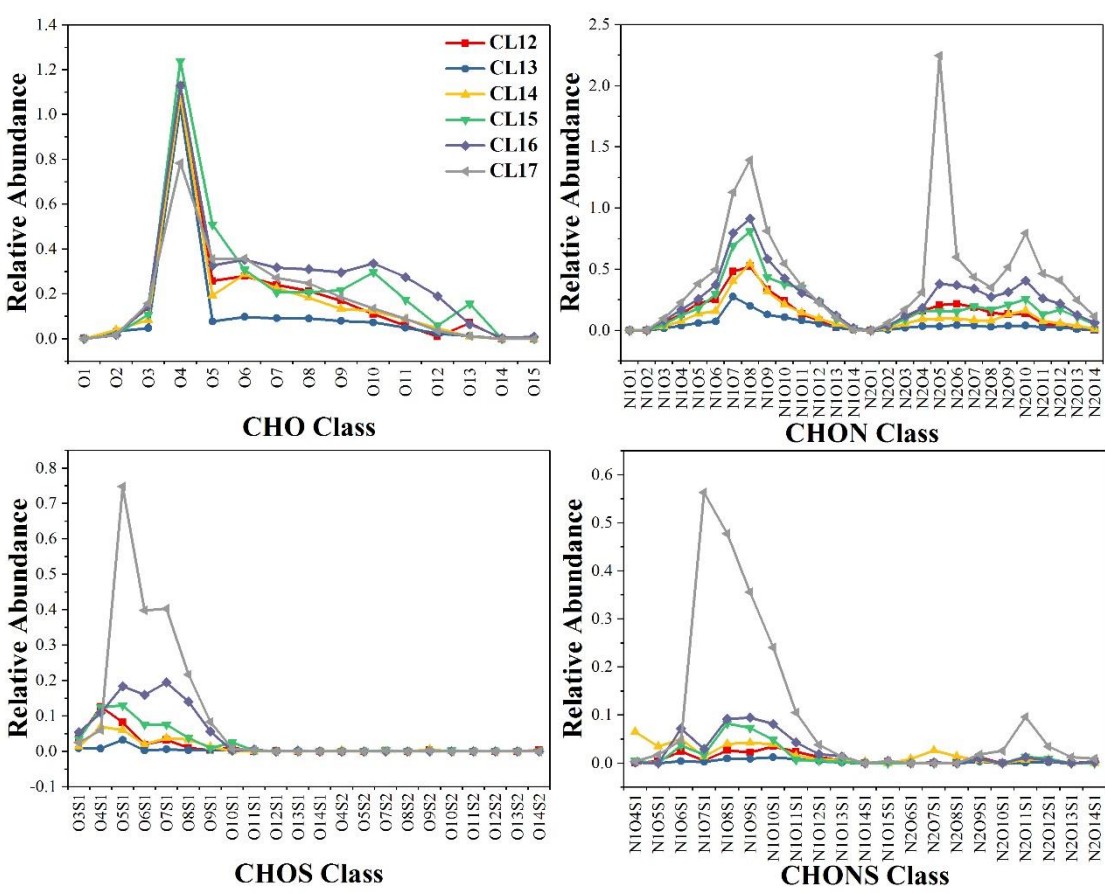

**Figure 4. Relative abundance of the categories of CHO, CHON, CHOS, and CHONS formulas according to the characteristic atom groups within the molecular formulas in cloud water.**

**Table 1**. **The sampling interval, liquid water content (LWC, g m$^{-3}$), and pH of each sample.**

| Time | Sample ID | Sampling Interval | LWC | pH |
|---|---|---|---|---|
| | CL12 | 2018/5/11 10:15-12:40 | 0.17 | 4.16 |
| Daytime | CL13 | 2018/5/11 12:40-15:00 | 0.17 | 4.22 |
| | CL14 | 2018/5/11 15:00-18:00 | 0.19 | 4.37 |
| | CL15 | 2018/5/11 18:00-21:00 | 0.17 | 4.28 |
| Nighttime | CL16 | 2018/5/11 21:00-24:00 | 0.16 | 4.18 |
| | CL17 | 2018/5/12 00:00-08:15 | 0.12 | 4.13 |