# Peer review of "Measurement report: Molecular characteristics of cloud water in southern China and insights into aqueous-phase processes from Fourier Transform Ion Cyclotron Resonance Mass Spectrometry"

_Atmospheric Chemistry and Physics, 2021_

## Referee Comment (RC1)

The work by Sun et al., presents the analysis of cloud water from Mt. Tianjing in southern China using Fourier transform-ion cyclotron resonance mass spectrometry (FT-ICR MS). The results are very interesting and provide some insight into in-cloud aqueous-phase chemical composition.

I have several comments and suggestions to the authors below.

Minor comments:

Considering the emphasis of the paper, I suggest moving sample preparation and analysis procedure into the main text (not the SI). In places it is not clear how the analysis was performed as incorrect referencing to the SI material section was provided. For example, SPE extraction was referred to Text S3, which does not contain this information, but described in the Text S1. *Please also see relevant comment below.*

The following sentence is not clear – ''The peaks of formulas are intensive within m/z 200-400'', line 139. Please rephrase. What is the impact of these observations?

Major comments:

It is not clear whether the authors used a direct infusion or applied a hyphenated technique for their analysis. This needs to be stated in the manuscript. Advantages and limitations of the applied technique need to be stated in the text as well.

The information on ESI parameters e.g.  source type, nebuliser gas pressure, gas velocity and temperatures, and capillary voltages are missing. These parameters are crucial to understand how the data was acquired and for a comparative analysis with the literature.  Major MS parameters have to be provided as well. How was the system calibrated **and** tuned? Depending on the system optimisation parameters the analyst would see preferentially one or other type of compounds.

What do the authors mean by the ''*mass spectra*'' calibration? ''*The mass spectra was calibrated externally using measurements of a known homologous series of N1 (neutral nitrogen compounds) and O2 compounds (acids) with high abundance in a petroleum*''. Please provide more details on the utilised petroleum type etc. If the system was optimised (tuned) using petroleum on N and O containing ''compounds'' it should not be a surprise that CHON, aliphatic and olefinic species are the major components in the analysed samples. I do not think it is correct to infer that N-containing compounds, aliphatic and olefinic species are the dominating organic species in the analysed water. It might be the case, but if your system is optimised for N-containing compounds then the analysis would be skewed towards these species. There is nothing wrong with this as there are no perfect analytical tools that would cover all compound classes; however, this needs to be acknowledged in the text, so that it is clear for the reader who are not expert in FTIR-MS.

In addition, there is a clear evidence that the type of SPE extraction (including Strata X cartridges) and ESI source (e.g. HESI, nano-ESI) can skew the recoveries of specific compound classes. It has been demonstrated that SPE sample pretreatment significantly improves ion recoveries for organic species with nonpolar and moderately polar functional groups, but leads to lower recoveries for highly oxygenated molecules. Therefore, while SPE reduced in-source adduct formation, it also limited the range of compounds identified through a single analysis (Kourtchev et al., 2020). So the observed variation of various molecular groups e.g. CHO, CHON presented in the work by Sun et al. work can be influenced by the applied techniques and thus lead to specific compound classes recoveries. How does this affect the interpretation of their data?

The authors identified and present elemental composition in their work but incorrectly refer (numerous times) to these formulae as compounds. This should be avoided as multiple isomers can be associated with a single molecular formula even at the reported by the author achieved <0.3ppm mass error especially for *mz* > 319.

**Reference:** Kourtchev et al. Comparison of Heated Electrospray Ionization and Nanoelectrospray Ionization Sources Coupled to Ultra-High-Resolution Mass Spectrometry for Analysis of Highly Complex Atmospheric Aerosol Samples, Anal. Chem. 2020, 92, 12, 8396–840, doi: 10.1021/acs.analchem.0c00971

---

## Author Comment (AC1)

**RESPONSE TO COMMENTS ON MANUSCRIPT #acp-2021-626**

**" Measurement report: Molecular characteristics of cloud water in southern China and insights into aqueous-phase processes from Fourier Transform Ion Cyclotron Resonance Mass Spectrometry" by Sun et al.**

We are grateful to the editor and three reviewers for their substantial efforts and helpful comments and suggestions, which are of great advantage to the improvement of the manuscript. The manuscript has been revised thoroughly according to the comments from three reviewers. Below, we detail responses and resulting edits to all of the reviewers' comments. We first list the review comments in normal font, then followed by our responses in blue. To make it clear, the contents in the revised manuscript are presented in quotes and in italics. References to line numbers are to the revised manuscript.

**Responses to comments by Referee 1**

General comments: The work by Sun et al., presents the analysis of cloud water from Mt. Tianjing in southern China using Fourier transform-ion cyclotron resonance mass spectrometry (FT-ICR MS). The results are very interesting and provide some insight into in-cloud aqueous-phase chemical composition.

Reply: Thanks for the reviewer's positive comments.

Minor comments:

Considering the emphasis of the paper, I suggest moving sample preparation and analysis procedure into the main text (not the SI). In places it is not clear how the analysis was performed as incorrect referencing to the SI material section was provided. For example, SPE extraction was referred to Text S3, which does not contain this information, but described in the Text S1. Please also see relevant comment below.

Reply: Thanks for the reviewer's suggestion. The contents about sample preparation and analysis procedure have been removed to the main text. Please refer to Lines 106-115 and 124-201. The other references to the

Supporting Information have also been carefully checked.

The following sentence is not clear – "The peaks of formulas are intensive within m/z 200-400'', line 139. Please rephrase. What is the impact of these observations?

Reply: Thanks for the comment. The sentence has been revised as "*The most intensive ion peaks are within the range of m/z 200−400.*" Please refer to Line 207. These observations indicated the distribution of molecular weight of assigned formulas, that is, species with molecular weights of ~200-400 Da are abundant in cloud water.

Major comments:

It is not clear whether the authors used a direct infusion or applied a hyphenated technique for their analysis. This needs to be stated in the manuscript. Advantages and limitations of the applied technique need to be stated in the text as well.

Reply: Thanks for the reviewer's helpful suggestion. The direct infusion method was used in this study. The samples were redissolved in 1 mL of methanol and injected into an electrospray ionization (ESI) source at a flow rate of 200 µL/h. We have clarified them in the revised manuscript, please refer to Lines 149-151.

Additionally, we have added the advantages and limitations of the applied technique as follows: "*ESI is a soft ionization technique that offers minimal fragmentation of the analytes (Mazzoleni et al., 2010). [M-H]⁻ was detected at the negative ion mode. The coupling of ESI and FT-ICR MS with ultra-high mass resolution makes it possible to characterize the element constitution within molecules. Note that ESI is efficient at ionizing molecules having polar functional groups containing nitrogen and oxygen atoms (Cho et al., 2015).*" , please refer to Lines 145-148, and "*Note that both the recovery of SPE and the selective ionization of negative ESI might cause a bias of mass spectra to certain peaks.*" , please refer to Lines 180-181.

**References**

Mazzoleni, L. R., B. M. Ehrmann, X. Shen, A. G. Marshall and J. L. Collett Jr (2010). "Water-Soluble Atmospheric Organic Matter in Fog: Exact Masses and Chemical Formula Identification by Ultrahigh-Resolution Fourier Transform Ion Cyclotron Resonance Mass Spectrometry." Environ. Sci. Technol. 44: 3690-3697.

Cho, Y., A. Ahmed, A. Islam and S. Km (2015). "Developments in FT-ICR MS instrumentation, ionization techniques, and data interpretation methods for petroleomics." Mass. Spectrom. Rev. 34(2): 248-263.

The information on ESI parameters e.g. source type, nebuliser gas pressure, gas velocity and temperatures, and capillary voltages are missing. These parameters are crucial to understand how the data was acquired and for a comparative analysis with the literature. Major MS parameters have to be provided as well. How was the system calibrated and tuned? Depending on the system optimisation parameters the analyst would see preferentially one or other type of compounds.

Reply: We agree with the reviewer's comments. In this study, a nebulizer gas pressure of 1 bar, a dry gas velocity of 4 L/min and temperature of 200 ℃, and capillary voltages of +4500 V and the end plate offset -500 V were used for ESI source (Bruker Daltonik GmbH, Bremen, Germany) at negative mode. The optimized mass for quadrupole (Q1) was 170 Da. An argon-filled hexapole collision pool was operated at 2 MHz and 1400 Vp-p RF amplitude. The time of flight was 0.7 ms and the mass range was 150-800 Da and the ion accumulation time was 0.1 s. This information has been added into Section 2.2, please refer to Lines 151-154.

What do the authors mean by the ''*mass spectra''* calibration? '*'The mass spectra was calibrated externally using measurements of a known homologous series of N1 (neutral nitrogen compounds) and O2 compounds (acids) with high abundance in a petroleum*''. Please provide more details on the utilised petroleum type etc. If the system was optimised (tuned) using petroleum on N and O containing ''compounds'' it should not be a surprise that CHON, aliphatic and olefinic species are the major components in the analysed samples. I do not think it is correct to infer that N-containing compounds, aliphatic and olefinic species are the dominating organic species in the analysed water. It might be the case, but if your system is optimised for N-containing compounds then the analysis would be skewed towards these species. There is nothing wrong with this as

there are no perfect analytical tools that would cover all compound classes; however, this needs to be acknowledged in the text, so that it is clear for the reader who are not expert in FTIR-MS.

Reply: Thanks for the constructive comments. Followed the methods described by Jiang et al. (2019), we used a known homologous series of $-N_1$ and $-O_2$ formulas (e.g., $C_{16}H_{31}O_2$, $C_{17}H_{33}O_2$, and $C_{18}H_{35}O_2$, etc. that only separated by $-CH_2$ units) frequently detected in a crude oil sample to calibrate the mass spectra before sample detection. The typical peaks (e.g., $-O_4$ species) in our samples were used to internally recalibrate the final spectrum. The calibration was used to improve the mass accuracy of the mass spectra. This information has been clarified in the revised manuscript, please refer to Lines 157-161.

The abundant CHO and CHON in our samples might be related to the bias of the ESI source at negative mode. Thus the acknowledgment of this issue has been added in the revised manuscript: " *Note that the abundant CHO and CHON cannot directly be related back to the composition of samples since the preferential detection of these molecules in negative ESI. However, the comparison among the samples is still meaningful since they are expected to have the same bias.*" Please refer to Lines 217-219.

**Reference**

Jiang, B., Z. W. Zhan, Q. Shi, Y. Liao, Y. R. Zou, Y. Tian and P. Peng (2019). "Chemometric Unmixing of Petroleum Mixtures by Negative Ion ESI FT-ICR MS Analysis." Anal. Chem. 91(3): 2209-2215.

In addition, there is a clear evidence that the type of SPE extraction (including Strata X cartridges) and ESI source (e.g. HESI, nano-ESI) can skew the recoveries of specific compound classes. It has been demonstrated that SPE sample pretreatment significantly improves ion recoveries for organic species with nonpolar and moderately polar functional groups, but leads to lower recoveries for highly oxygenated molecules. Therefore, while SPE reduced in-source adduct formation, it also limited the range of compounds identified through a single analysis (Kourtchev et al., 2020). So the observed variation of various molecular groups e.g. CHO, CHON presented in the work by Sun et al. work can be influenced by the applied techniques and thus lead to specific compound classes recoveries. How does this affect the interpretation of their data?

Reply: We agree with the reviewer's comments that both the SPE procedure and ESI source have selectivity for the analytes. The pretreatment of SPE in this study mainly followed the methods in studies that focused on molecular characteristics of cloud water (e.g., Zhao et al., 2013, Cook et al., 2017, Bianco et al., 2018). The operation of SPE was different from Kourtchev et al. (2020), in which the neutral organic compounds were eluted with 0.1% formic acid in methanol. While the mixed solution of acetonitrile/methanol/water (45/45/10, *v*:*v*:*v*) at pH 10.4 was used in this study to elute the analytes. The recovery was not evaluated in our study. However, the Strata-X (Phenomenex) cartridges with both hydrophilic and hydrophobic functional groups are expected a high recovery (Zhao et al., 2013).

As the reviewer mentioned above, selectivity cannot be excluded. However, the high oxygenated molecules ($O/C > 1$), which were demonstrated to have a low recovery (Kourtchev et al., 2020), were not the situation in our study since we have set the selecting criteria of $O/C \leq 1.2$ to exclude formulas undetected frequently in natural materials. Moreover, all the samples were pretreated and detected in the same procedure; thus the same bias was expected. Therefore, the comparison among the samples is still meaningful.

We have clarified the possible selectivity of SPE procedures for analytes in Section 2: "*Note that both the recovery of SPE and the selective ionization of negative ESI might cause a bias of mass spectra to certain peaks.*" Please refer to Lines 180-181. The caveats for the results were also added to the revised manuscript: "*Note that the abundant CHO and CHON cannot be directly related back to the composition of samples since the preferential detection of these molecules in negative ESI. However, the comparison among the samples is still meaningful since they are expected to have the same bias.*" Please refer to Lines 217-219.

**References**

Zhao, Y., A. G. Hallar and L. R. Mazzoleni (2013). "Atmospheric organic matter in clouds: exact masses and molecular formula identification using ultrahigh-resolution FT-ICR mass spectrometry." Atmos. Chem. Phys. 13(24): 12343-12362.

Cook, R., Y.-H. Lin, Z. Peng, E. Boone, R. K. Chu, J. E. Dukett, M. J. Gunsch, W. Zhang, N. Tolic, A. Laskin and K. A. Pratt (2017). "Biogenic, urban, and wildfire influences on the molecular composition of dissolved organic compounds in cloud water." Atmos. Chem. Phys. 17(24): 15167-15180.

Bianco, A., L. Deguillaume, M. Vaitilingom, E. Nicol, J. L. Baray, N. Chaumerliac and M. Bridoux (2018). "Molecular Characterization of Cloud Water Samples Collected at the Puy de Dome (France) by Fourier Transform Ion Cyclotron Resonance Mass Spectrometry." Environ. Sci. Technol. 52(18): 10275-10285.

Kourtchev, I., P. Szeto, I. O'Connor, O. A. M. Popoola, W. Maenhaut, J. Wenger and M. Kalberer (2020). "Comparison of Heated Electrospray Ionization and Nanoelectrospray Ionization Sources Coupled to Ultra-High-Resolution Mass Spectrometry for Analysis of Highly Complex Atmospheric Aerosol Samples." Anal. Chem. 92(12): 8396-8403.

The authors identified and present elemental composition in their work but incorrectly refer (numerous times) to these formulae as compounds. This should be avoided as multiple isomers can be associated with a single molecular formula even at the reported by the author achieved <0.3 ppm mass error especially for $mz > 319$.

Reply: Thanks for the helpful comments. We have replaced the word "compounds" with "molecules" or "formulas" in the revised manuscript.

**Responses to comments by Referee 2**

General Comments: The manuscript by Sun et al. presents the mass spectral characteristic of cloud water samples throughout a long-lasting cloud event by FT-ICR-MS, and attempts to shed light on the potential influences of in-cloud aqueous phase reactions, which are currently uncertain for the formation of SOA. They show that CHON with aromatic structures are the most abundant type in cloud water, suggesting their enhanced formation in cloud. Their results also indicate distinctly differences between day and night, which is most probably attributed to diurnal differences in aqueous chemistry. Such observation could provide valuable cloud chemistry data for the community, and has the potential to be published after considering my comments. The major weakness is the limited dataset, thus the authors have to clearly indicate in the discussion of the diurnal difference of cloud chemistry between day and night, since there could be other factors contributing to such difference.

Reply: We agree with the reviewer's comments that the dataset is limited in our study. We took the reviewer's

suggestion and revised Section 3.4.1 by summarizing the diurnal difference of cloud chemistry as follows: *"Previous studies have revealed the differences in atmospheric chemistry between day and night. The daytime chemistry is dominated by photochemical reactions, in which OH radical oxidation and photolysis represent the main processes in the aqueous phase (Ervens et al., 2011). While during the nighttime, $NO_3$ radical is dominant (Herrmann et al., 2010). The radical nitration of phenols by $NO_2$ and $NO_3$ radicals leads to the formation of nitrophenols (Harrison et al., 2005)."* Please refer to Lines 351-355.

Other factors, including liquid water content, pH value of cloud water, and the meteorological condition during sampling, were stable as we described in Supporting Information. Nevertheless, the influences of other factors cannot be excluded. We thus clarified that in Section 4: *"We noted that the database for the diurnal variation analysis is limited in this study, but the results provided novel insights into the diurnal variation of cloud chemistry. Firm conclusions warrant future field studies."* Please refer to Lines 399-400.

**References**

Ervens, B., B. J. Turpin and R. J. Weber (2011). "Secondary organic aerosol formation in cloud droplets and aqueous particles (aqSOA): a review of laboratory, field and model studies." Atmos. Chem. Phys. 11(21): 11069-11102.

Herrmann, H., D. Hoffmann, T. Schaefer, P. Brauer and A. Tilgner (2010). "Tropospheric aqueous-phase free-radical chemistry: radical sources, spectra, reaction kinetics and prediction tools." Chemphyschem 11(18): 3796-3822.

Harrison, M. A. J., S. Barra, D. Borghesi, D. Vione, C. Arsene and R. Iulian Olariu (2005). "Nitrated phenols in the atmosphere: a review." Atmos. Environ. 39(2): 231-248.

Specific Comments:

-Introduction: Overall it is OK, but it would be better to include the aqueous formation mechanisms related to CHON and CHOS.

Reply: Thanks for the reviewer's helpful suggestion. We included the aqueous-phase formation mechanisms of CHON, mainly including the radical nitration and carbonyls-ammonium/amine reactions, in the original

manuscript. Additional information about the aqueous formation of organonitrates was added as follows: "*In addition, the nucleophilic addition of nitrate to the isoprene-derived epoxydiol can effectively form the organonitrates (Darer et al., 2011).*" Please refer to Lines 86-87. The new information about the formation mechanisms of organosulfates was added: "*Several formation mechanisms of organosulfates, such as acid-catalyzed ring-opening of epoxides, sulfate esterification, nucleophilic substitution of alcohols with sulfuric acid, and sulfoxy radical reactions, have been proposed in recent years (Brüggemann et al., 2020).*" Please refer to Lines 88-91.

**References**

Darer, A. I., N. C. Cole-Filipiak, A. E. O'Connor and M. J. Elrod (2011). "Formation and stability of atmospherically relevant isoprene-derived organosulfates and organonitrates." Environ. Sci. Technol. 45(5): 1895-1902.

Bruggemann, M., R. Xu, A. Tilgner, K. C. Kwong, A. Mutzel, H. Y. Poon, T. Otto, T. Schaefer, L. Poulain, M. N. Chan and H. Herrmann (2020). "Organosulfates in Ambient Aerosol: State of Knowledge and Future Research Directions on Formation, Abundance, Fate, and Importance." Environ. Sci. Technol. 54(7): 3767-3782.

- Lines 172, "the current understanding that aqueous-phase reactions generally increase the degree of oxidation (Ervens et al., 2011)." Please also include the reasons to this understanding. Does such aqueous reactions refer to in cloud processing?

Reply: Thanks for the reviewer's comments. In the aqueous phase, the precursors and products of aqueous-phase reactions generally exhibit higher water-solubility and polarity than those in the gas phase. The enhanced formation of SOA with a high oxidation degree in the aqueous phase has been observed in many studies (Ge et al., 2012). Here the aqueous phase refers to both aerosol liquid water and cloud droplets. The related sentence has been revised to "*This is not consistent with the current understanding that precursors and products in the aqueous phase have a higher O/C, which generally causes the high water-solubility of molecules (Ervens et al., 2011).*" Please refer to Lines 243-244.

**References**

Ervens, B., B. J. Turpin and R. J. Weber (2011). "Secondary organic aerosol formation in cloud droplets and aqueous particles (aqSOA): a review of laboratory, field and model studies." Atmos. Chem. Phys. 11(21): 11069-11102.

Ge, X., Q. Zhang, Y. Sun, C. R. Ruehl and A. Setyan (2012). "Effect of aqueous-phase processing on aerosol chemistry and size distributions in Fresno, California, during wintertime." Environ. Chem. 9(3): 221-235.

- Lines 182, "The O/C ratios and OSC of CHO collected during the daytime is slightly lower than the nighttime…". What about the influence of primary emission? Since the samples collected during the daytime and nighttime may originally presents different characteristics without oxidation.

Reply: We agree with the reviewer's comment that the primary emissions might also affect the differences between daytime and nighttime samples. The sampling site was mainly influenced by long-distance transport rather than local emissions. The wind direction and the air masses origin during sampling did not change dramatically. However, the 72-hour back trajectory of air masses showed that more continental air masses might be included during the daytime than the nighttime (Fig. S1). Thus the influence of air masses origin and the aging processes cannot be excluded. However, being limited by the sample size, the firm conclusion is difficult to draw. Thus the following sentence is added to the revised manuscript: "*The difference of air masses' origin and the aging processes may also influence the cloud chemistry. However, since the database is limited, the further conclusion cannot be drawn based on them.*" Please refer to Lines 261-262.

- Lines 195, is there any result of aromaticity related to traffic emission or other sources, in addition to coal combustion and biomass burning? Since the present OA molecular does not correspond to these sources, i.e., coal combustion and biomass burning as discussed.

Reply: Thanks for the comments. The fraction of aromatic structures in the WSOC of traffic emission aerosols is also high (> 30% for CHO and CHON, and >20% for CHOS) (Tang et al., 2020). To clarify the question

more persuasively, we here additionally compared the results of our study with the reports on urban aerosols, which are mainly influenced by local primary emissions. The related sentences have been revised and the new texts have been added: "*However, it is quite different from the primary emissions, including biomass burning, coal combustion, and traffic emission, of which the fraction of aromatic structures is higher (Song et al., 2018; Tang et al., 2020). The urban aerosols collected in Guangzhou, southern China, which may be mainly influenced by local primary emissions, also have a high fraction of aromatic molecules (> 20%) (Zou et al., 2020), implying the aging processes likely reduce the aromaticity of organics.*" Please refer to Lines 276-280.

**References**

Tang, J., J. Li, T. Su, Y. Han, Y. Mo, H. Jiang, M. Cui, B. Jiang, Y. Chen, J. Tang, J. Song, P. Peng, and G. Zhang (2020). "Molecular compositions and optical properties of dissolved brown carbon in biomass burning, coal combustion, and vehicle emission aerosols illuminated by excitation-emission matrix spectroscopy and Fourier transform ion cyclotron resonance mass spectrometry analysis." Atmos. Chem. Phys. 20(4): 2513-2532.

Song, J., M. Li, B. Jiang, S. Wei, X. Fan and P. Peng (2018). "Molecular Characterization of Water-Soluble Humic like Substances in Smoke Particles Emitted from Combustion of Biomass Materials and Coal Using Ultrahigh-Resolution Electrospray Ionization Fourier Transform Ion Cyclotron Resonance Mass Spectrometry." Environ. Sci. Technol. 52(5): 2575-2585.

Zou, C., M. Li, T. Cao, M. Zhu, X. Fan, S. Peng, J. Song, B. Jiang, W. Jia, C. Yu, H. Song, Z. Yu, J. Li, G. Zhang and P. a. Peng (2020). "Comparison of solid phase extraction methods for the measurement of humic-like substances (HULIS) in atmospheric particles." Atmos. Environ. 225: 117370.

- Lines 251, It is an interesting result that coal combustion contributes to S-containing formulas in cloud water more significantly compared with CHO and CHON. Is there any other evidence to support the demonstration, such as the correlation between CHOS with the concentration of SO$_2$ or sulfate?

Reply: Thanks for the constructive comments. We conducted correlation analysis between CHOS with the

concentration of $SO_2$ and sulfate, and found no statistical correlation between CHOS and $SO_2$ ($p > 0.05$) but significant correlation for CHOS and $SO_4^{2-}$ ($r^2 = 0.72$, $p < 0.05$) in cloud water. However, since the database for the regression analysis was too limited (n = 6), the result might be subject to considerable uncertainty. Thus we did not include the result in the manuscript.

- Lines 278, "For CHO, the most abundant $C_{17}H_{26}O_4$ in cloud water is not detected in the $PM_{2.5}$ samples, suggesting a formation by the in cloud aqueous-phase reactions, although the contribution from BVOCs cannot be ruled out." Reasons should be discussed for such a contradiction.

Reply: Thanks for the comments. The sentences in the original manuscript may be amphibolous. As we discussed in Section 3.3, $C_{17}H_{26}O_4$ was detected in α-pinene ozonolysis SOA. However, $C_{17}H_{26}O_4$ was detected in cloud water but not in $PM_{2.5}$ in this study, indicating that it may mainly form through in-cloud aqueous-phase reactions at this sampling site. We have revised the sentence as follow: "*For CHO, the most abundant $C_{17}H_{26}O_4$ in cloud water was also detected in α-pinene ozonolysis SOA as we discussed in Section 3.3. However, it was not detected in $PM_{2.5}$ in this study, indicating that it may mainly form through in-cloud aqueous-phase reactions.*", please refer to Lines 365-367.

**Responses to comments by Referee 3**

This work provides a comprehensive analysis of the compounds in cloud water as well as the interstitial PM2.5, and characterize the distribution of different groups of species by using FT-ICR-MS; daytime and nighttime comparison was also made. Based on such analysis, aqueous-phase processing and the reactions involved were inferred to enhance our understanding of the aerosol chemistry. The paper is fairly well written and provide useful information and knowledge regarding the cloud water organics, this reviewer however has a series of comments to be addressed first before its acceptance.

Reply: We would like to thank the reviewer for his/her positive comments.

Line 51: as you stated here, "chromatographic and spectroscopic techniques only determined ~20% of all kinds of organics", then what does the FT-ICR-MS perform? Even though it has a super high mass resolution, is it being able to determine all existing species? If not, organics with what functionalities are preferred to be detected? What is the fraction of determined species to the total? How does this bias affect your interpretation? I think this issue should be clarified in your manuscript.

Reply: Thanks for the reviewer's comments. The detection of ESI FT-ICR MS has a certain selectivity; thus it cannot detect all the existing species. ESI is efficient at ionizing molecules having polar functional groups containing nitrogen and oxygen atoms, but not for molecules lacking nitrogen or oxygen atoms (Cho et al., 2015). Moreover, the reduced nitrogen is not easy to be detected at the negative ion mode. So the related sentence has been revised as "*Ultra-high resolution mass spectrometry such as Fourier Transform Ion Cyclotron Resonance Mass Spectrometry (FT-ICR MS) has made it possible to provide more comprehensive information of individual molecular formulas in complex mixtures, although the selectivity of detection still exists (Cho et al., 2015; Hockaday et al., 2009).*" Please refer to Line 56. The texts about the selectivity of ESI have been added in Section 2.2: "*ESI is efficient at ionizing molecules having polar functional groups containing nitrogen and oxygen atoms (Cho et al., 2015).*" "*Note that both the recovery of SPE and the selective ionization of negative ESI might cause a bias of mass spectra to certain peaks.*" Please refer to Lines 148-149 and 180-181.

Up to our best knowledge, there are no analytical tools that can cover all compound classes, so the fraction of determined species to the total is challenging to evaluate. Hocladay et al. (2009) estimated that 13% of the dissolved organic matter is either undetected or underrepresented by combined ESI (+,-) and APPI (+) (atmospheric pressure photoionization ionization) sources. Although the ionization of ESI source is selective, which might cause a bias of the distribution of molecular composition to some extent, the same bias was expected since all the samples were pretreated and detected in the same procedure. We have clarified them in the revised manuscript: "*Note that the abundant CHO and CHON cannot directly be related back to the composition of samples since the preferential detection of these molecules in negative ESI. However, the comparison among the samples is still meaningful since they are expected to have the same bias.*" Please refer to Lines 217-219.

**References**

Cho, Y., A. Ahmed, A. Islam and S. Kim (2015). "Developments in FT-ICR MS instrumentation, ionization techniques, and data interpretation methods for petroleomics." Mass. Spectrom. Rev. 34(2): 248-263.

Hockaday, W. C., J. M. Purcell, A. G. Marshall, J. A. Baldock and P. G. Hatcher (2009). "Electrospray and photoionization mass spectrometry for the characterization of organic matter in natural waters: a qualitative assessment." Limnol. Oceanogr.: Meth. 7: 81-95.

Section 2.2: Even though instrumental details are included in the supplement, I think some key information, for example, the mass resolution, and how to remove background organics, etc can be briefly described here, as well as the IC and TOC/TN analysis.

Reply: Thanks for the reviewer's suggestion. The detailed information about instrument analysis has been removed into the main text, please refer to Lines 106-115 and 124-201.

As described in (2), background organics or impurities during sample storage and treatment, might be detected as FT-ICR-MS is highly sensitive and has ultrahigh resolution. The number of molecules in cloud samples seem to be much higher than those in PM2.5 samples, I am wondering how do these excess compounds come from except from possible aqueous-phase processing?

Reply: Thanks for the comments. In our study, we made blank samples for the high-quality data in order to draw safe conclusions. The results we provided in this study have been corrected by blank samples. Thus the impact of background organics or impurities is considered to be minor. We detected 1264-2767 formulas in cloud water and 1057-1198 formulas in $PM_{2.5}$ samples. The number of assigned formulas may be mainly related to the concentration of organics in extracts of samples. Since the cloud water and $PM_{2.5}$ samples have different concentrations, it is not surprising that the different number of formulas were detected. Therefore, we used the relative fraction (e.g., the fraction of four groups to the total), the statistical results (e.g., average O/C ratios), and some formulas with huge differences in relative abundance between cloud water and $PM_{2.5}$ to do comparison and to indicate the impacts of in-cloud reactions. We have clarified that in the revised

manuscript: *"The smaller number of assigned formulas in PM$_{2.5}$ may be mainly related to the low concentration of total organics in PM$_{2.5}$ extracts."* Please refer to Lines 221-222.

Line 170-179: This reviewer thinks that cloud cycling might need to be considered, as the interstitial PM2.5 sampled here may contain aqueous oxidation products inside cloud droplets as cloud droplets in reality cycle per few minutes. Therefore a high O/C value might be observed in PM2.5 samples rather than cloud water. Whether or not aqueous processing could enhance the oxidation degree of organics depends on the ageing time. In a short time scale, the organic oxidation degree could increase and the more oxidized species may fragment into low oxygenated ones given enough time.

Reply: We are grateful to the reviewer for providing his/her constructive views. Previous studies using the large-eddy simulation model have shown that the parcel in-cloud residence time is on the scale of a few minutes (Stevens et al., 1996; Feingold et al., 1998). These studies mainly focused on the stratocumulus in the boundary layer with turbulence. However, the sampling site in this study was located at the top of a mountain, and the meteorological conditions at the observation site were stable during sampling. The temperature ranged from 15.2 to 15.9°C, and the relative humidity was stable at 100%, which is unfavorable for the droplets' evaporation. Moreover, the wind direction did not change dramatically, which may imply a minor impact of the turbulence. If the interstitial aerosols underwent several cloud cycles during the sampling, the composition of cloud water and interstitial aerosols would tend to be consistent. However, the previous studies have revealed the differences between cloud residues and interstitial aerosols (e.g., Roth et al., 2016; Lin et al., 2017). Thus we infer that the impact of cloud cycling would be limited before the cloud event ending although we cannot completely rule out their influence. The related texts have been added: *"Previous studies using the large-eddy simulation model have shown that the in-cloud residence time of the parcel is on the scale of a few minutes (Stevens et al., 1996; Feingold et al., 1998), thus some masses formed in cloud droplets may remain in aerosols via the evaporation of the droplets, resulting some high oxidation organics entering the interstitial PM$_{2.5}$. We cannot completely rule out the influence of cloud cycling, however, this impact may be limited because of the stable meteorological conditions with constant temperature, wind and saturated or supersaturated water vapor during sampling (Fig S2)."* Please refer to Lines 246-251.

**References**

Stevens, B., Feingold, G., Cotton, W. R., & Walko, R. L. (1996). "Elements of the Microphysical Structure of Numerically Simulated Nonprecipitating Stratocumulus. " J. Atmos. Sci., 53(7): 980-1006.

Feingold, G., Kreidenweis, S. M., and Zhang, Y. (1998). "Stratocumulus processing of gases and cloud condensation nuclei: 1. Trajectory ensemble model. " J. Geophys. Res., 103(D16): 19527-19542.

Roth, A., J. Schneider, T. Klimach, S. Mertes, D. van Pinxteren, H. Herrmann and S. Borrmann (2016). "Aerosol properties, source identification, and cloud processing in orographic clouds measured by single particle mass spectrometry on a central European mountain site during HCCT-2010." Atmos. Chem. Phys. 16(2): 505-524.

Lin, Q., G. Zhang, L. Peng, X. Bi, X. Wang, F. J. Brechtel, M. Li, D. Chen, P. Peng, amp, apos, an, G. Sheng and Z. Zhou (2017). "In situ chemical composition measurement of individual cloud residue particles at a mountain site, southern China." Atmos. Chem. Phys. 17(13): 8473-8488.

Line 286-292: Similar for OS, typically OS can be produced more efficiently in aerosol water rather than liquid water, yet no statistical difference are observed here, as there are repeated cycling between cloud water and interstitial PM2.5.

Reply: Thanks for the reviewer's helpful comments. As we mentioned above, the impact of cloud cycles would be minor before the cloud event ending in the case of this study. However, we clarified that the organosulfates may form in aerosol efficiently: "*For CHOS formulas, the most abundant functions classes are similar between cloud water and $PM_{2.5}$. No statistical difference of the fraction of organosulfates is observed between cloud water and $PM_{2.5}$ except for a low $f_{RA}$ (69.5%) of organosulfates in P2 sample, which may indicate the wide variety of formation mechanisms (e.g., acid-catalyzed particle-phase reactions, nucleophilic substitution reactions in aqueous phase) and/or other common sources of CHOS in cloud water and $PM_{2.5}$ (Bruggemann et al., 2020), but possible slightly enhanced formation of that in cloud water.*" please refer to Lines 374-378.

**Reference**

Bruggemann, M., R. Xu, A. Tilgner, K. C. Kwong, A. Mutzel, H. Y. Poon, T. Otto, T. Schaefer, L. Poulain, M.

N. Chan and H. Herrmann (2020). "Organosulfates in Ambient Aerosol: State of Knowledge and Future Research Directions on Formation, Abundance, Fate, and Importance." Environ. Sci. Technol. 54(7): 3767-3782.

Specific comments

Line 25-26, do you mean CHON and CHO-containing species? It is not clear. The last sentence in Line 26 is not a full sentence.

Reply: Yes, CHON and CHO here mean formulas containing C, H, O, N and C, H, O respectively. The sentence has been revised as "*CHON (formulas containing C, H, O, and N elements, the same is true for CHO and CHOS) represents the dominant component (43.6-65.3% of relative abundance), followed by CHO (13.8-52.1%).*" The last sentence has been revised as "*S-containing formulas constitute ~5-20% of all assigned formulas.*" Please refer to Lines 26-28.

Line 28: A recent paper by Wang et al (2021, 118:e2022179118) demonstrates that aqueous-phase oxidation of aromatic species could be a source of SOA, this might be a supporting evidence that "CHON with aromatic structures are abundant in cloud water"

Reply: Thanks for the reviewer's helpful suggestions. The citation has been added in Line 284: "*The aromatic species may provide the precursors of aqueous-phase reactions (Wang et al., 2021).*"

**Reference**

Wang, J., J. Ye, Q. Zhang, J. Zhao, Y. Wu, J. Li, D. Liu, W. Li, Y. Zhang, C. Wu, C. Xie, Y. Qin, Y. Lei, X. Huang, J. Guo, P. Liu, P. Fu, Y. Li, H. C. Lee, H. Choi, J. Zhang, H. Liao, M. Chen, Y. Sun, X. Ge, S. T. Martin and D. J. Jacob (2021). "Aqueous production of secondary organic aerosol from fossil-fuel emissions in winter Beijing haze." Proc. Natl. Acad. Sci. U.S.A. 118(8): e2022179118.

Line 69: Consider to add citation Ye et al., Atmos Environ 2020;223:117240, which determines the organic acids produced from aqueous-phase oxidation of a certain precursor.

Reply: Thanks for the reviewer's kind suggestion. The citation has been added to the revised manuscript, please refer to Line 72.

Line 144: RA means relative abundance, this reviewer somehow think the authors can directly use "relative abundance", it is easier to understand than RA.

Reply: Thanks for the reviewer's suggestion. The word "RA" has been replaced by "relative abundance" in the revised manuscript.